# High Throughput Priority-Based Layered QC-LDPC Decoder with Double Update Queues for Mitigating Pipeline Conflicts

**DOI:** 10.3390/s22093508

**Published:** 2022-05-05

**Authors:** Yunfeng Li, Yingchun Li, Nan Ye, Tianyang Chen, Zhijie Wang, Junjie Zhang

**Affiliations:** Key Laboratory of Specialty Fiber Optics and Optical Access Networks, Joint International Research Laboratory of Specialty Fiber Optics and Advanced Communication, Shanghai University, Shanghai 200444, China; lyf1282@shu.edu.cn (Y.L.); liyingchun@shu.edu.cn (Y.L.); aslanye@shu.edu.cn (N.Y.); tyral_chen@shu.edu.cn (T.C.); zhijie_wang@shu.edu.cn (Z.W.)

**Keywords:** double update queues, high throughput, pipeline conflicts, QC-LDPC, priority-based

## Abstract

A high-throughput layered decoder for quasi-cyclic (QC) low-density parity-check (LDPC) codes is required for communication systems. The preferred way to improve the throughput is to insert pipeline stages and increase the operating frequency, which suffers from pipeline conflicts at the same time. A priority-based layered schedule is proposed to keep the updates of log-likelihood ratios (LLRs) as frequent as possible when pipeline conflicts happen. To reduce pipeline conflicts, we also propose double update queues for layered decoders. The proposed double update queues improve the percentage of updated LLRs per iteration. Benefitting from these, the performance loss of the proposed decoder for the fifth generation (5G) new radio (NR) is reduced from 0.6 dB to 0.2 dB using the same quantization compared with the state-of-the-art work. As a result, the throughput of the proposed decoder improved up to 2.85 times when the signal-to-noise ratio (SNR) was equal to 5.9 dB.

## 1. Introduction

As a forward error correction (FEC) code, low-density parity-check (LDPC) code [1] has an excellent performance close to the Shannon limit. Due to its exploitable parallelism, LDPC decoder was easily implemented on field-programmable gate array (FPGA) devices [2].

Among various LDPC codes, a quasi-cyclic (QC) LDPC code exhibits high facility in structural routing and memory addressing. QC-LDPC code has a parity-check matrix (PCM) that is composed of all-zero submatrices and circularly shifted identity submatrices. PCM can be simply represented by a base graph matrix. Owing to these advantages, QC-LDPC is now widely applied in many communication standards (ITU [3], DVB-S2X [4], WiMAX [5], and 5G NR [6]).

With the increase in communication rate, LDPC decoder, as a key component, needs to increase the throughput. The methods involve reducing the iteration number together with the decoding cycles per iteration and increasing the operating frequency [7].

In order to decrease the number of iterations, the layered decoding schedule [8] is widely applied since it shows twice decoding convergence for the same decoding performance compared with the flooding decoding schedule [9].

The decoding cycle per iteration can be reduced by improving the processing parallelism. Considering the tradeoff between the throughput and hardware utilization, the common approach is to use the partially parallel architecture [10]. The parallelism is usually equal to the lifting size (Z), which is the size of submatrix in PCM.

Higher operating frequency can be achieved by inserting more pipeline stages. Nevertheless, this will increase the probability and number of pipeline conflicts. Pipeline conflicts will impact the update of log-likelihood ratios (LLRs), result in the loss of decoding performance, and increase extra iteration numbers for the same decoding performance.

### 1.1. Related Works

To address the pipeline conflicts, various solutions have been proposed. The conventional solution is to insert additional stall cycles and wait for a conflict-free pipeline. Pipeline conflicts can be eliminated by adjusting not only the processing order of submatrices in the layer but also the processing order of layers [11]. By adopting this method, [12] a throughput of 1.2 Gbps at eight iterations in 5G NR is achieved. Nevertheless, pipeline conflicts still occur frequently by using this method in a relatively dense base graph matrix. Ref. [13] proposes to split the layer with the size equal to Z into several smaller layers to reduce the occurrence of pipeline conflicts while the throughput is lowered as well. The residue-based layered schedule [14] postpones the update of LLR and stores the contributions of decoding into registers when a pipeline conflict occurs. However, there exists an extreme circumstance that LLR of a variable node may never be updated when pipeline conflicts always happen to it in a dense base graph matrix. This significantly degrades the performance of the layered schedule. In [15], the flooding schedule is adopted when pipeline conflicts occur in a layered decoder, so it is called a hybrid decoder. In [16], an improved normalized probabilistic min-sum algorithm (INPMSA) was proposed to compensate the decline in the decoding performance. In the check node unit (CNU), the probabilistic second minimum is revised by using the first minimum and proportion fixing.

### 1.2. Overview and Contribution

In this paper, we focus on mitigating pipeline conflicts at a high operating frequency in a layered decoding schedule. This is achieved by the following contributions:

(1) Double update queues replace the single update queue in the layered LDPC decoder. In comparison with [15], the percentage of up-to-date LLR read operations per iteration with double update queues during the decoding was increased by up to 31%.

(2) The priority-based layered decoding schedule is proposed to update LLRs as frequent as possible when the pipeline conflicts cannot be avoided. Due to a higher percentage of updated LLRs and more frequent update of LLRs, the priority-based decoder with double update queues lower performance loss from 0.6 dB to 0.2 dB compared with [15]. Comparing with residue-based decoder [14] using double update queues, the proposed decoder shows an advantage of 0.1 dB.

(3) As a direct result of the proposed layered LDPC decoder, the throughput of our proposed decoder for 5G NR is up to 2.85 times that of [15] with the same quantization on the Xilinx VC709 evaluation board at the same signal-to-noise ratio (SNR).

The remaining sections are organized as follows. In Section 2, the layered decoding schedule is introduced. In Section 3, the proposed priority-based layered decoding schedule and double update queues are described. The structure of the proposed priority-based layered decoder with double update queues and the flow chart of the proposed decoder are also described. In Section 4, the results of simulation and hardware implementation are presented. Finally, Section 5 provides a conclusion to this paper.

## 2. Layered Decoding Schedule

LDPC code is decoded based on the iterative message-passing algorithm, which means the decoding messages are exchanged frequently between the check nodes and variable nodes. The decoding messages include variable-to-check message, check-to-variable message, and a posterior probability LLR (APP-LLR). In the layered decoding schedule, LLRs are commonly updated after the update of variable-to-check messages and check-to-variable messages in a layer.

For convenience of presentation, we make the following definitions. $V_{v,c}^{it}$ denotes the variable-to-check messages that propagated from the variable node *v* to the check node *c* at the *it*-th iteration. $R_{c,v}^{it}$ denotes the check-to-variable messages that propagated from the check node *c* to the variable node *v* at the *it*-th iteration. LLR corresponding to the variable node *v* in the *it*-th iteration is represented as $LLR_{v}^{it}$.

Before the start of decoding, APP-LLR from an additive white Gaussian noise (AWGN) channel is initialized as given by
(1)$$LLR_{v}^{init}=log\frac{P\left(x_{v}=0|y_{v}\right)}{P\left(x_{v}=1|y_{v}\right)}=\frac{2y_{v}}{\sigma^{2}},$$
where $P\left(x_{v}=0|y_{v}\right)$ represents the probability that $x_{v}$ is equal to 0 and $P\left(x_{v}=1|y_{v}\right)$ represents the probability that $x_{v}$ is equal to 1. The received signal from the channel is represented as $y_{v}$ and channel noise variance is represented as $\sigma^{2}$.

In the *it*-th iteration, the variable-to-check messages are generated with the check-to-variable messages from the previous iteration and LLR as given by
(2)$$V_{v,c}^{it}=LLR_{v}^{it}-R_{c,v}^{it-1}.$$

In the hardware implementation, the min-sum algorithm (MSA) [17] is employed for the update of check-to-variable messages because of the friendly implementation. It turns the complexity computations into the simple comparison operations at the cost of decoding performance degradations. MSA only needs to select two minimum messages from check nodes. The calculation of MSA is given by
(3)$$R_{c,v}^{it}=\underset{v^{\prime }\in V_{c}\backslash v}{\prod }sgn\left(V_{v^{\prime },c}^{it}\right)\cdot \underset{v^{\prime }\in V_{c}\backslash v}{min}\left(\left|V_{v^{\prime },c}^{it}\right|\right),$$
where $V_{c}$ denotes the group of variable nodes that connected to the check node *c* and $V_{c}\backslash v$ represents the same group of variable nodes except the variable node *v*.

In order to improve the decoding performance, two improved min-sum algorithms were proposed in [18]. They use a correction factor to correct the magnitude of the two minimum values in the check-to-variable update. The offset min-sum algorithm (OMSA) subtracts a correction factor β from the two minimum values. The normalized min-sum algorithm (NMSA) uses a correction factor α to multiply the two minimum values. The OMSA and NMSA are calculated as given by the two following equations
(4)$$R_{c,v}^{it}=\alpha \underset{v^{\prime }\in V_{c}\backslash v}{\prod }sgn\left(V_{v^{\prime },c}^{it}\right)\cdot \underset{v^{\prime }\in V_{c}\backslash v}{min}\left(\left|V_{v^{\prime },c}^{it}\right|\right)$$
(5)$$R_{c,v}^{it}=\underset{v^{\prime }\in V_{c}\backslash v}{\prod }sgn\left(V_{v^{\prime },c}^{it}\right)\cdot max\left(\underset{v^{\prime }\in V_{c}\backslash v}{min}\left(\left|V_{v^{\prime },c}^{it}\right|\right)-\beta ,0\right).$$

The APP-LLR of the variable node *v* is updated as the following equation
(6)$$LLR_{v}^{it}=V_{v,c}^{it}+R_{c,v}^{it}$$

At the end of an iteration, the codeword *C* is decided based on the value of APP-LLR. If the APP-LLR of variable node *v* is no less than zero, the bit will be decided to be 0. If the APP-LLR of variable node *v* is negative, the bit will be decided to be 1. The decided codeword *C* and the parity-check matrix *H* then generate the syndrome $S=C\times H^{T}$. Suppose that the LDPC decoder considers an early termination and the number of iterations is limited in the hardware implementation. There exist two cases to terminate the decoding. One case is that the syndrome is equal to zero and the decoding has not reached the set maximum iteration number. The other case is that the syndrome is not equal to zero when the iteration number has reached the set maximum iteration number.

## 3. Priority-Based Layered QC-LDPC Decoder with Double Update Queues

### 3.1. Priority-Based Layered Decoding Schedule

In the layered schedule, the LLR in a layer has not been updated yet while the next layer needs this updated LLR. This is called pipeline conflict. When a pipeline conflict occurs, the practical method is to ignore the update of LLR. However, a small percentage of ignored updates will lead to significant performance degradation [19]. For this reason, we propose a priority-based layered schedule.

In the layered schedule, the update of LLR can be equivalent to the sum of LLR and difference between the newly calculated check-to-variable messages in the current iteration and the one in the previous iteration [15]. This difference can be understood as a gain that helps the decoding. The update of LLR can be expressed as (7)
(7)$$LLR_{v}^{L_{i},it}=LLR_{v}^{L_{i-1},it}+R^{L_{i},it}-R^{L_{i},it-1}\;=LLR_{v}^{L_{i-1},it}+G_{v}^{L_{i},it},$$
where $LLR_{v}^{L_{i},it}$ represents the LLR for the variable node *v* at the *it*-th iteration in the $L_{i}\;$ layer, $R^{L_{i},it}$ represents the check-to-variable messages at the *it*-th iteration in the $L_{i}$ layer and $G_{v}^{L_{i},it}$ represents the gain for the variable node *v* at the *it*-th iteration in the $L_{i}$ layer.

In the priority-based layered schedule, when a pipeline conflict happens to two check nodes between two adjacent layers, $L_{i}$ and $L_{i+1}$, new LLR will be updated in the layer $L_{i}$. The gain $G^{L_{i+1},it}$ will be calculated in the layer $L_{i+1}$. The gain $G^{L_{i+1},it}$ then will be added to the newly updated LLR later. In this way, updates of LLRs can be guaranteed no matter how the base graph matrix is dense.

Suppose there are three check nodes in layers $L_{i}$, $L_{j}$ and $L_{k}$ connected to the same variable nodes. During decoding, pipeline conflicts happen between $L_{i}$ and $L_{j}$, $L_{j}$ and $L_{k}$.

The priority-based layered schedule works as follows. The LLR in layer $L_{i}$ can update in priority and get $LLR_{v}^{L_{i},it}$. Due to pipeline conflicts, the layer $L_{j}$ reuses the old LLR value $LLR_{v}^{L_{i-1},it}$ to calculate the gain $G_{v}^{L_{j},it}$. If the update of LLR in layer $L_{i}$ can be done before decoding the layer $L_{k}$, then the layer $L_{k}$ can update the value based on the result of $LLR_{v}^{L_{i},it}$ and the gain $G_{v}^{L_{j},it}$ can also be added to the updated LLR in layer $L_{k}$. As shown in Figure 1, the update can be expressed as (8) and (9)
(8)$$LLR_{v}^{L_{i},it}=V^{L_{i},it}+R^{L_{i},it}$$
(9)$$LLR_{v}^{L_{k},it}=V^{L_{k},it}+R^{L_{k},it}+G_{v}^{L_{j},it}$$
where the variable-to-check message at the *it*-th iteration in the layer $L_{k}$ is denoted as $V^{L_{k},it}$.

### 3.2. Structure of the Priority-Based Layered LDPC Decoder with Double Update Queues

Figure 2 shows the detailed architecture of priority-based layered LDPC decoder with double update queues. The parallelism of processing units is equal to the size of submatrix in the corresponding PCM. In the WiMAX decoder, the parallelism is equal to 96 [5]. In the 5G NR decoder, the parallelism is equal to 384 [6].

Before decoding, LLRs are initialized and denoted as $LLR^{itit}$s. They are stored into the LLR RAM. LLR RAM is composed of a simple dual-port block RAM (BRAM) which is used to store the latest updated LLR. The old LLR value can be read repeatedly provided that no new LLR value is written in the same address. This feature is conductive to the implementation of the proposed decoding schedule.

When the decoding starts, LLR is read out from RAM according to the address given and sent to the LLR barrel shifter. It is not essential to use the reverse barrel shifter to shuffle the submatrix as the identity matrix before storing back to the LLR RAM [20]. Instead, the barrel shifter that shuffles based on the absolute shift value can be well applied in the decoder. After shuffling, LLR is sent into variable node units (VNUs) to calculate $V^{it}$s. Then, $V^{it}$s are passed to the CNUs. At the same time, $V^{it}$s are buffered into FIFOs waiting for the update of LLR.

In the CNUs, the minimum (min) and the second minimum (smin) absolute value, the index of the minimum value (min index), and the sign product (sign product) of $V^{it}$s are achieved. Then, registers will store this intermediate data until new $R^{it}$s are required for update in the next layer. $R^{it}$s generated from CNU are added with $V^{it}$ buffered out from FIFO and achieve the updated LLRs. When LLRs are updated, they are stored back to the LLR RAM.

In this paper, we propose double update queues instead of a single update queue [21] to update the LLRs. Compared with a single update queue, double update queues accelerate the update of LLRs in a layer and decrease the occurrence of pipeline conflicts. This will be discussed in detail in Section 3.3.

When pipeline conflict happens, the gain calculator module and gain adder module are used to migrate the conflicts. These will also be discussed in detail combining with the flow chart in Section 3.3.

Due to the design of priority-based schedule and double update queues, the signs of all the $V^{it}$s are buffered into four separated FIFOs in the CNU. The two minimum values, the sign product and the index of the minimum are used to generate the $R^{it-1}s$ for VNU used in the next iteration and $R^{it}$s for overlapping submatrices, non-overlapping variable nodes (double update queues) and gain (priority-based schedule).

### 3.3. Double Update Queues

There are two reasons for pipeline conflicts in the layered decoder. The first reason is that the inserted pipeline stages result in the highly delayed update to LLRs. When increasing the operating frequency by inserting more pipeline stages, it will inevitably lead to conflict probability. The second reason is the failure to buffer the variable-to-check message out from FIFO and add it to the corresponding check-to-variable message to obtain the updated LLR when the next layer needs this LLR. In order to address this problem, it is necessary to increase the flexibility of data being buffered into FIFO and buffering out from FIFO. If variable-to-check messages in one layer can be stored in several FIFOs separately, then variable-to-check messages can be buffered out in time for the update of LLRs when the next layer needs them. In this way, pipeline conflicts can be eliminated. However, this consumes plenty of memory resources.

To trade off the memory resources and the possibility of pipeline conflicts, we proposed the double update queues. In the double update queues, we use two FIFOs to buffer variable-to-check messages. One FIFO is called overlapping FIFO. The other FIFO is called non-overlapping FIFO. Overlapping FIFO is used to buffer those variable-to-check messages whose LLRs will continue to be decoded in the next layer. A non-overlapping FIFO is used to buffer variable-to-check messages whose LLRs will not be needed in the next layer. Note that if an LLR of a submatrix can be updated regularly in the current layer and would suffer pipeline conflicts in the next layer, the variable-to-check message of this submatrix in the current layer will be buffered into the non-overlapping FIFO and its updated LLR will be written back to the LLR RAM. The variable-to-check message of this submatrix in the next layer will not be buffered into any FIFOs because the corresponding LLR cannot be updated. FIFOs for buffering the signs need to be divided into overlapping and non-overlapping FIFOs as well. Two separate queues to generate new $R^{it}$s in CNU are also needed. In each update queue, variable-to-check messages are added with check-to-variable message to achieve their updated LLR separately. In this way, LLRs can be updated in double queues.

Combined with the priority-based schedule and double update queues, we introduce the decoding flow chart in detail as shown in Figure 3.

In our design, variable nodes in a layer are processed in units of submatrix. For simplicity of presentation, the variable nodes in a submatrix are denoted as variable node group (VNG). Before decoding, the processing order of VNGs in a layer needs to be reordered. In a layer, the VNGs that have not been decoded in the previous layer are decoded first. Next are the VNGs that have been decoded in the previous layer.

When the processing order of VNGs is determined, decoding starts. LLRs are successively read out from LLR RAM. After the processing of barrel shifter and VNU, variable-to-check messages are obtained. According to the mechanism of double update queues, variable-to-check messages are buffered into overlapping FIFO or non-overlapping FIFO. Then, the check-to-variable messages are updated.

The next step is the process of the LLR update when the pipeline conflict occurs or does not occur. If no pipeline conflicts happen, LLRs can be normally updated as the layered schedule. After update, LLRs of non-overlapping VNGs will be written back to the LLR RAM. LLRs of overlapping VNGs will be bypassed to the barrel shifter and participate in the decoding in the next layer. At the end of one iteration, the codeword will be decided according to the sign of LLRs. If the iteration has reached the maximum iteration number or the calculated syndrome is equal to zero, the decoding will end. If not, the decoding will continue.

If pipeline conflicts happen during the decoding, LLRs of the VNG with conflicts will not be updated. Combining with Figure 2, the impact of pipeline conflicts on decoding can be mitigated as follows. If the LLR in the previous layer has not been updated yet, then the old LLR value is read again from LLR RAM for the current layer. VNU calculates its variable-to-check message $V^{it}$ and passes it to the CNU. At the same time, $V^{it}$ is not necessary to be buffered into FIFO because its corresponding LLR will not be updated in this layer. Different from the layered schedule, $R^{it}$ is calculated separately with sign buffered in a separate FIFO. Then, it will minus the $R^{it-1}$ obtained from the previous iteration and obtain the gain $G^{it}$. Before storing the gain $G^{it}$ into RAM, gain $G^{it}$ should enter the gain barrel shifter and be shuffled to the corresponding position of the submatrix that it will be added with in the other layer. When the LLR is updated in other layers like in the layered decoding schedule, the gain $G^{it}$ then adds to this updated LLR.

### 3.4. Detailed Illustration of the Proposed Decoder with High Performance

To help understand the mechanism of the priority-based layered decoder with double update queues, here we give an example. The timing diagram of decoding with the QC-LDPC code PCM is shown in Figure 2. RD address and WR address represent the addresses that LLR reads from and writes to. All the addresses are unified with the index of VNGs. Double update queues work as follows. During decoding, overlapping FIFO buffers variable-to-check messages $V^{it}$s of overlapping VNGs that their newly updated LLRs will participate decoding in the next layer. Bypass means those LLRs will be bypassed to the barrel shifter instead of being written back into LLR RAM. Non-overlapping FIFO buffers $V^{it}$s of non-overlapping VNGs that their newly updated LLRs will be written to memory. LLRs of overlapping and non-overlapping VNGs are updated separately once their respective check-to-variable messages are calculated.

As shown in Figure 4, the base graph matrix is dense and has four rows and nine columns. The number of pipeline stages is set to three. When an LDPC code is being decoded in a conventional layered decoder, VNGs in a layer are processed as the order shown in PCM. When a pipeline conflict occurs, stall cycles are necessarily inserted to maintain the full decoding performance. As an example, in the first layer, the first, second, fourth, fifth, and sixth VNGs participate in the decoding in order. In the second layer, the second, third, fourth, sixth, and seventh VNGs participate in the decoding in order. At the ninth cycle shown in Figure 4, LLRs of variable nodes in the first layer are written back to RAM in sequence. To avoid pipeline conflicts, five stall cycles have to be inserted and LLRs of the second VNG in the second layer cannot be read out from RAM until the 11th cycle, since the updated LLRs of the second VNG in the first layer are written back to the RAM at the 10th cycle.

The residue-based layered decoder, hybrid decoder, and priority-based layered decoder with double update queues eliminate stall cycles so that LLRs of a VNG can be read out from memory at each cycle. The solution to pipeline conflicts in the residue-based decoder [14] works as follows. At the sixth cycle, there exists a pipeline conflict to the second VNG. LLRs of the second VNG have to read the old LLR values from the RAM and use these values for decoding. At the 10th cycle, the gain of the second VNG in the first layer is saved in a register file for patching. The second VNG in the second layer can be updated normally at the 14th cycle and the gain is added with the updated LLRs when the LLR write operation happens, here referred as patched LLR write. In this way, the performance loss is compensated. However, the residue-based decoder has to postpone updates of LLRs when the pipeline conflicts happen to the LLRs in one variable node [14]. In this example, LLRs in the fourth VNG can never be updated because of the pipeline conflict and postponed patch.

In the hybrid decoder, the solution for pipeline conflicts works as follows. In the first layer, updated LLRs of the second, fourth, and sixth VNGs are written to both the LLR memory and FIFO (double write) [15]. The patched LLR update of the second, fourth, and sixth VNGs is done as shown in Equation (7) at the 14th, 16th, and 17th cycle, respectively. In this manner, LLR updates are not postponed and check node gains are added as soon as they are ready. However, the number of the occurrence of pipeline conflicts is still high.

In our proposed priority-based decoder with double update queues, the processing of the decoding is shown in detail in Figure 3. Before the start of decoding, the processing order of VNGs is needed to be reordered. As shown in Figure 4, in the first layer, the fifth, and sixth VNGs are first decoded since they are not decoded in the fourth layer in the previous iteration. Then, the first, second, and fourth VNGs are decoded. In the second layer, the third, seventh, second, fourth, and sixth VNGs are decoded in turn. In the third layer, the first, fifth, eighth, seventh, and fourth VNGs are decoded in turn. In the fourth layer, the second, ninth, first, eighth, and fourth VNGs are decoded in turn.

After the LLRs are read from memory, they are used to calculate the variable-to-check messages. In the first layer, the variable-to-check messages of the fifth, first, and second VNGs are buffered into the non-overlapping FIFO since LLRs of the fifth and first VNGs will not participate in the decoding in the second layer and the updated LLRs of the second VNG will not be used in the second layer. In the first layer, the variable-to-check messages of the sixth and fourth VNGs are buffered into overlapping FIFO since these variable nodes are needed in the second layer after their LLRs are updated. The VNGs of other layers also buffer in this way. After the update of LLRs, LLRs of the overlapping submatrices are bypassed to the data path. They continue to be decoded in the next layer. LLRs of the non-overlapping variable nodes are written back to the memory. In the first layer, the fifth, first, and second VNGs are non-overlapping. Their LLRs are written back to memory. On the contrary, LLRs of the non-overlapping sixth and fourth VNGs are bypassed to the data path and participate in the decoding in the second layer. In this way, LLRs are updated in double queues and the occurrence of pipeline conflicts is obviously decreased.

When a pipeline conflict happens, the solution in the priority-based decoder works as follows. According to the priority-based schedule, LLRs of the second VNG in the first layer have priority to update at the 11th clock. At the eighth cycle, a pipeline conflict happens to the second VNG in the second layer. Therefore, it has to read the old LLR values because the LLR values of the second VNG in the first layer have not been updated yet at the eighth cycle. The variable-to-check messages of the second VNG in the second are calculated and passed to CNU but not buffered into overlapping FIFO or non-overlapping FIFO. The second VNG in the second layer calculates the gain on the basis of variable-to-check messages and stores the gain. At the 24th cycle, LLRs of the second VNG in the fourth layer are first updated normally after the occurrence of the pipeline conflict. At this moment, the gain of the second VNG in the second layer is added to the updated LLRs of the second VNG in the fourth layer. In this way, the loss caused by the pipeline conflict is compensated.

## 4. Hardware Implementation and Result Discussion

### 4.1. Verification of Pipeline Conflict Reduction for Double Update Queues

From the illustration shown in Section 3.4, it can be seen that double update queues can reduce the pipeline conflicts and increase the percentage of updated LLRs during decoding. To demonstrate the effect of double update queues in making more LLRs updated, we choose the PCM in 5G NR (code rate = 22/27). Figure 5 shows the percentage of updated LLRs per iteration during decoding depending on the number of pipeline stages. With the increase in pipeline stages, the percentage of update LLRs per iteration is gradually getting worse. Compared with optimized results in [15], the double update queues increase the percentage of updated LLRs by 4–31%. Compared with the single update queue, the improvement is between 12% and 63%.

### 4.2. Analysis of the Decoding Performance

In order to directly reflect the effectiveness of the priority-based schedule and double update queues in improving decoding performance, we made a Monte Carlo simulation to obtain the frame error rate (FER) curves in the AWGN channel as shown in Figure 6. One million codewords are sent for each SNR. The maximum iteration number was set to 10 and the modulation format was set to quad-phase shift keyed (QPSK). The PCM is chosen from 5G NR with code rate 22/27. The simulated priority-based decoder with a single update queue, priority-based decoder with double update queues, residue-based decoder with a single update queue, and residue-based decoder with double update queues are all with 13 pipeline stages. All these four decoders were implemented as an offset min-sum decoder with LLRs quantized to eight bits and messages quantized to six bits, as [15] did.

In order to compare the decoding performance fairly and reflect the decoding performance accurately, we take FER = $10^{-5}$ as the standard as [15] did. From Figure 6, it is apparent to see that double update queues significantly improve the decoding performance. The priority-based decoder with double update queues shows a gain of 0.4dB compared with the one with a single update queue. The residue-based decoder with double update queues achieves a gain of 6.5dB compared with the one with a single update queue, since some LLRs of variable nodes can never be updated with a single update queue during decoding. From Figure 6, it can also be found that the priority-based decoder needs lower SNR than the residue-based decoder when achieving the same decoding performance because it updates LLRs more frequently.

Another Monte Carlo simulation was also done in the AWGN channel for 5G NR (code rate 22/27) and WiMAX (code rate 3/4) to show the results between the SNR and frame error rate (FER) of various decoders, as shown in Figure 7. In the Monte Carlo simulation, one million codewords are sent out to calculate the FER for each SNR. The simulation for 5G NR was performed for different maximum iteration numbers (*it_max_* = 10, *it_max_* = 20 and *it_max_* = 30). The simulation for WiMAX was performed when the maximum iteration number was set to 10. For a fair comparison with [15], the number of pipeline stages in 5G NR was set to 13, as [15] did. The number of pipeline stages in WiMAX was set to 10. The detail of the hardware implementation will be discussed in Section 4.3. The algorithm and quantization of LLRs and messages were also set as [15] did, where the algorithm was OMSA, the LLRs are quantized as eight bits and messages are quantized as six bits. The modulation format was set to QPSK.

For a fair comparison with the hybrid decoding, we take FER = $10^{-5}$ as the standard as [15] did. In Figure 7a, it can obviously be seen that for 5G NR the loss of SNR performance between the layered decoder and the priority-based decoder with double update queues is 0.2 dB. The loss of SNR performance between the layered decoder and the hybrid decoder is 0.6 dB. Thus, the decoding performance loss at FER = $10^{-5}\;$ narrowed from 0.6 dB to 0.2 dB by using the priority-based layered schedule with double update queues when the maximum iteration is set to 10. When the maximum iteration is set to 20 and 30, the loss of SNR performance does not exist. In order to reflect the improvement of the double update queues, performance of residue-based layered decoder with double update queues is also simulated. The loss of the residue-based decoder is just 0.3 dB after using the double update queues at 10 iterations. Note that the priority-based layered schedule has a faster convergence than the residue-based layered schedule. In Figure 7d, for WiMAX, the loss of SNR performance between the layered decoder and the priority-based decoder with double update queues is only 0.1 dB when the iteration number is set to 10. For WiMAX, the loss of SNR performance between the layered decoder and the residue-based decoder with double update queues is 0.2 dB. The simulation results for WiMAX also shows the effect of priority-based decoder with double update queues in reducing the loss of decoding performance caused by pipeline conflicts.

Analysis of average iteration number among different decoders is shown in Figure 8. The maximum iteration number is set to 30. The iteration finishes when codeword *C* and PCM *H* satisfy $C\times H^{T}=0$ or the decoding has reached the maximum number of iterations. From Figure 8, it can be found that at the same SNR, the average iteration number of the priority-based schedule with double update queues is highly reduced compared with that of the hybrid schedule. This greatly improves the throughput of the decoder. At higher SNR, the iteration of our design is nearly half of the hybrid layered decoder.

### 4.3. Hardware Implementation

The implementation results of our decoders and previous works are shown in Table 1 in detail. In this table, we use {$LLR_{v_{i}}^{init}$, $LLR_{v_{i}}^{it}$, $R_{j,i}^{it}$} to define the quantization as [12] did. For a fair comparison, numbers of pipeline stages for 5G NR and WiMAX decoders were set to 13 and 10 as [15] did. During the calculation, all LLRs and messages were subject to overflow processing. In the decoder, OMSA was used to calculate the check-to-variable messages and offset factor was set to 0.125. The normalized throughput *T_norm_* represents the throughput for one decoding iteration. From Table 1, it is obvious to see that *T_norm_* of decoder we designed is as high as previous works. The double update queues bring a little complexity in routing. As a result, the maximum frequency the decoders can operate at is a little lower than [15] but still high enough to provide a high throughput. Our decoders consume a bit more logic resources than [12,15]. The consumed look-up tables (LUTs), flip-flops (FFs), and BRAMs account for only 24%, 10%, and 7% of the xc7vx690t, respectively.

Figure 9 exhibits the resource usage of every module in the hardware implementation of the priority-based decoder with double update queues. In our decoders, only the LLR RAM, gain RAM, and check-to-variable FIFO are built with 36k BRAMs. Other FIFOs and buffers are built with Distributed RAMs (DRAMs). Thus, the 36k BRAMs used in our decoders are much less than other decoders. Although the design of double update queues seems to consume a lot of resources, the LUTs used for double update queues account for only 15.5% of the LUTs in the decoder. The FFs used for double update queues account for only 15.4% of the FFs in the decoder.

### 4.4. Analysis of Throughput

According to the results of average iteration number and *T_norm_* in Section 4.2 and Section 4.3, the throughput ratio between the priority-based decoder and hybrid decoder [15] is exhibited in Table 2. In Table 2, *AIN* represents the average iteration number and *T* represents the throughput of decoders. *TR* represents the throughput ratio between the priority-based layered decoder with double update queues and the hybrid layered decoder. To meet all the practical applications, *AIN* s are rounded up, slightly different with the results shown in Figure 6. From Table 2, it can be seen that *TR* ranges from 158% to 285%.

## 5. Conclusions

In this paper, we have proposed: (1) a priority-based layered schedule, enabling LLRs to be frequently updated when pipeline conflicts occur, and (2) double update queues that separately update LLRs of overlapping and non-overlapping submatrices, for reducing pipeline conflicts. The increase in percentage of updated LLRs per iteration is up to 31% compared with the state-of-the-art work. Therefore, the performance loss decreases from 0.6dB to 0.2dB. The throughput rises to 2.85 Gbps when the SNR is equal to 5.9dB.

Considering that the consumed LUTs, FFs, and 36k BRAMs only account for 24%, 10%, and 7% of the FPGA device xc7vx690t, respectively, for one QC-LDPC decoder core, it is expected that a higher throughput can be obtained easily through a multi-core architecture. Certainly, a higher-end FPGA device for UltraScale+ series has more resources and can embed more LDPC decoder cores. A 13-core LDPC decoder with four iterations can achieve a throughput beyond 100 Gbps at 6.9dB. The multi-core decoder will be implemented and verified on the UltraScale+ FPGA board in the real-time communication systems in the future work.

## Figures and Tables

**Figure 1 sensors-22-03508-f001:**
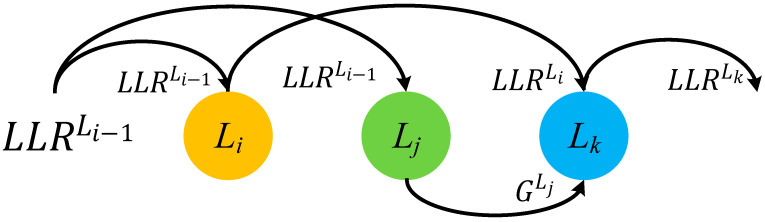
Illustration of the priority-based layered schedule for pipeline conflicts.

**Figure 2 sensors-22-03508-f002:**
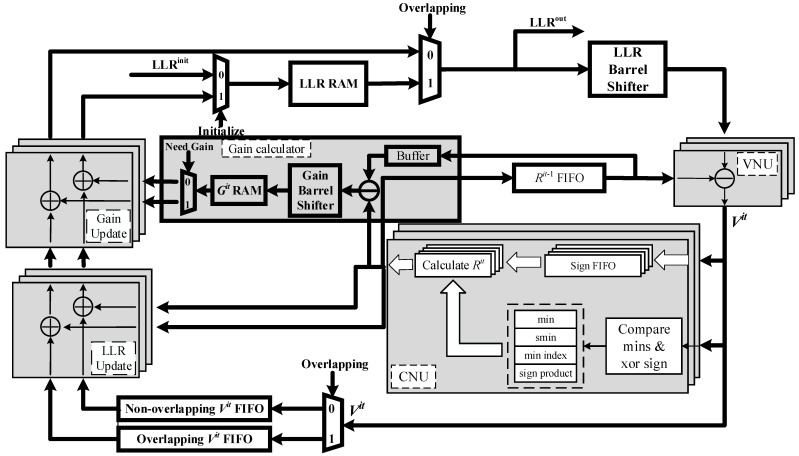
Architecture of the priority-based QC-LDPC decoder with double update queues.

**Figure 3 sensors-22-03508-f003:**
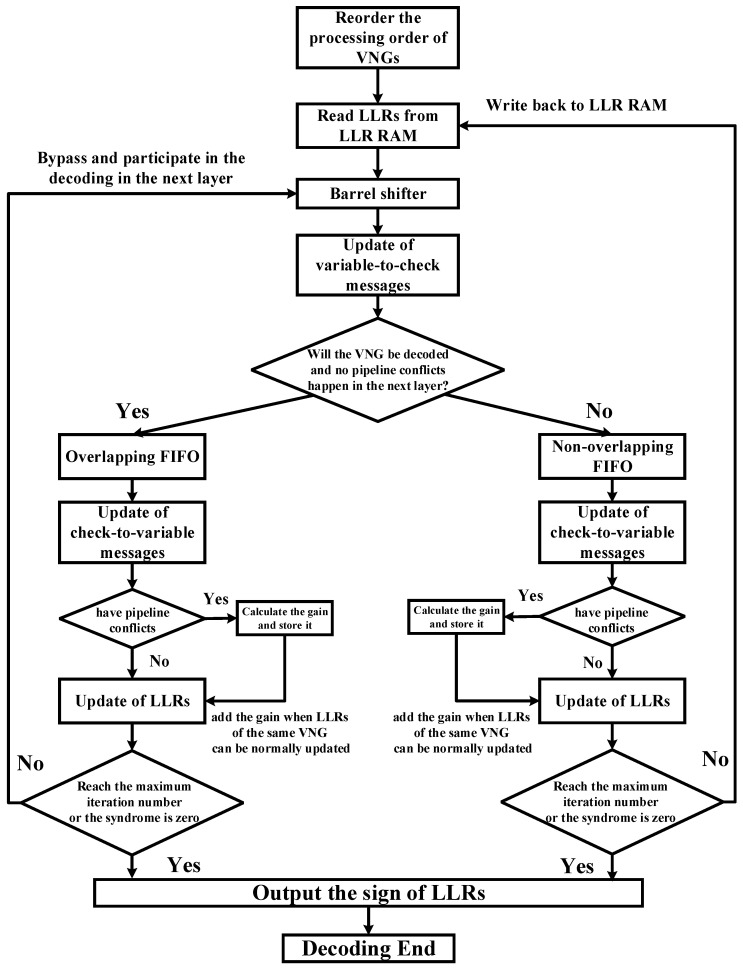
The flow chart of the priority-based layered schedule.

**Figure 4 sensors-22-03508-f004:**
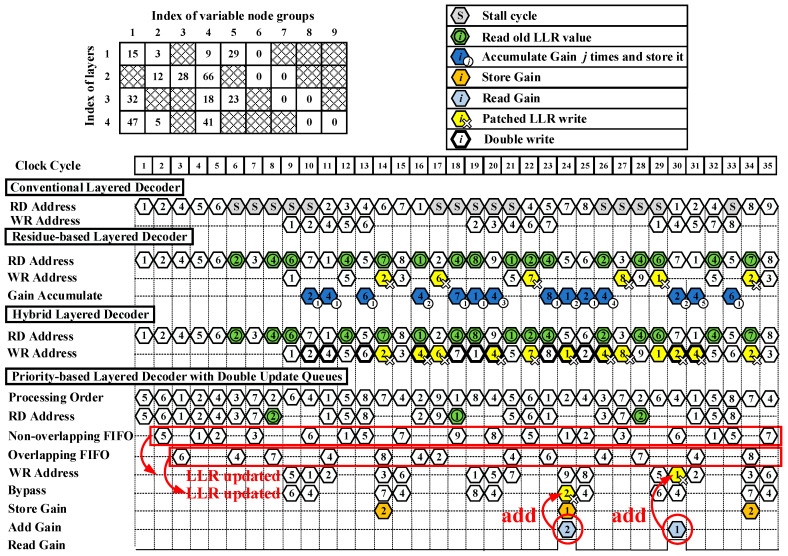
Illustration of the priority-based layered decoder with double update queues compared with the conventional layered decoder, the residue-based decoder and the hybrid decoder. The number of pipeline stages is set to 3.

**Figure 5 sensors-22-03508-f005:**
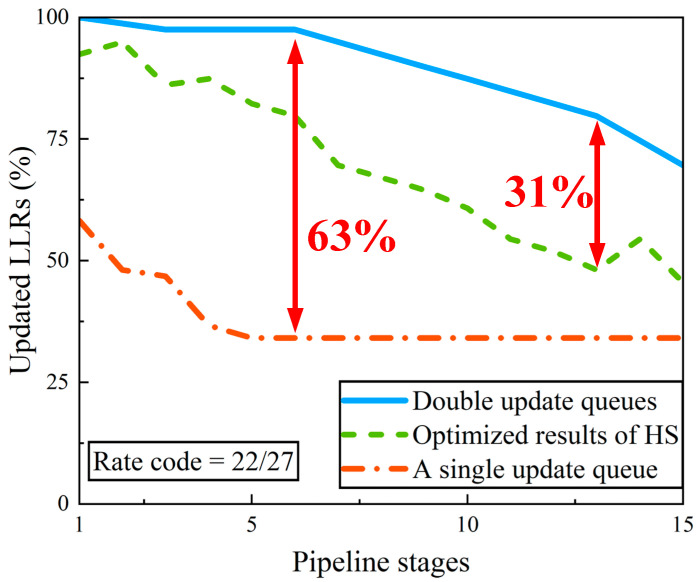
Percentage of updated LLRs per iteration during decoding as a function of number of pipeline stages for rate 22/27 from 5G NR. HS: Decoder with the hybrid schedule.

**Figure 6 sensors-22-03508-f006:**
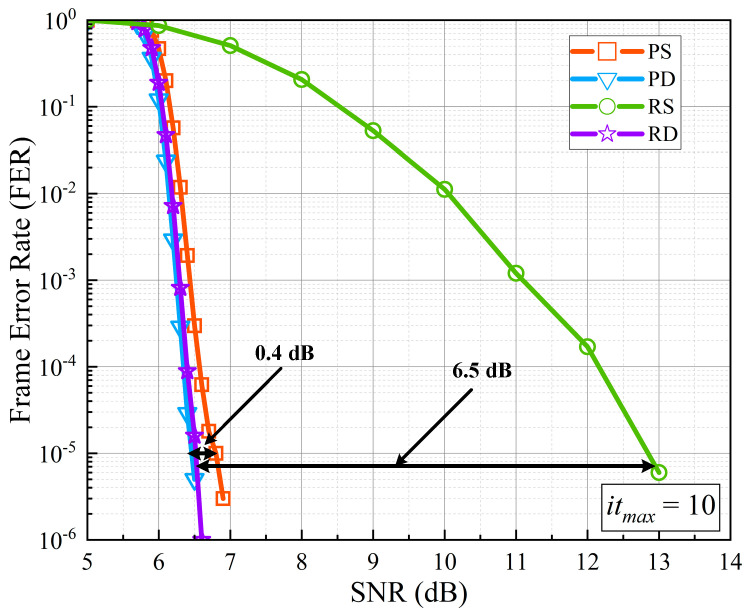
The SNR performance of different schedules. PS: Priority-based decoder with a single update queue. PD: Priority-based decoder with double update queues. RS: Residue-based decoder with a single update queue. RD: Residue-based decoder with double update queues.

**Figure 7 sensors-22-03508-f007:**
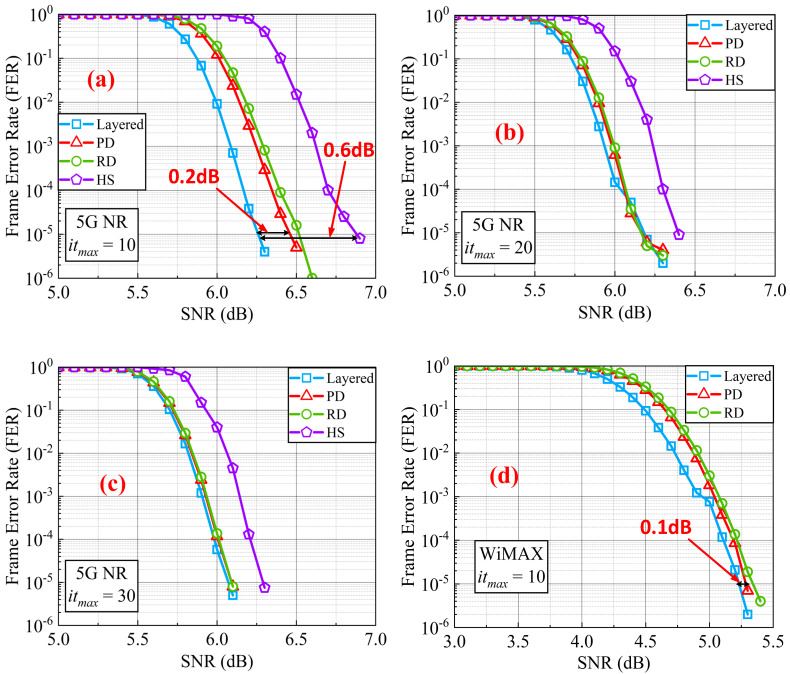
The SNR performance of different schedules with different maximum iteration numbers. PD: Priority-based decoder with double update queues. RD: Residue-based decoder with double update queues. HS: Decoder with the hybrid schedule. (**a**) the maximum iteration number is set to 10 for 5G NR. (**b**) the maximum iteration number is set to 20 for 5G NR. (**c**) the maximum iteration number is set to 30 for 5G NR. (**d**) the maximum iteration number is set to 10 for WiMAX.

**Figure 8 sensors-22-03508-f008:**
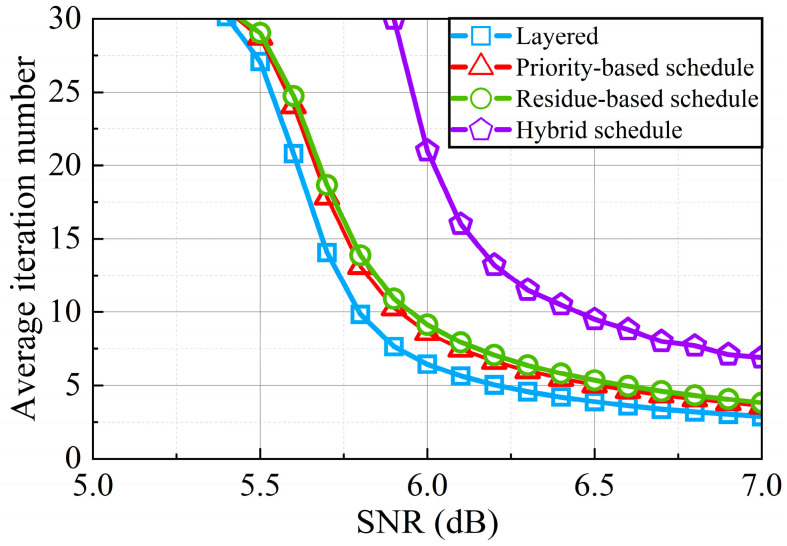
Average iteration number necessary for successful decoding for 5G NR (code rate = 22/27) compared with the result of the hybrid schedule.

**Figure 9 sensors-22-03508-f009:**
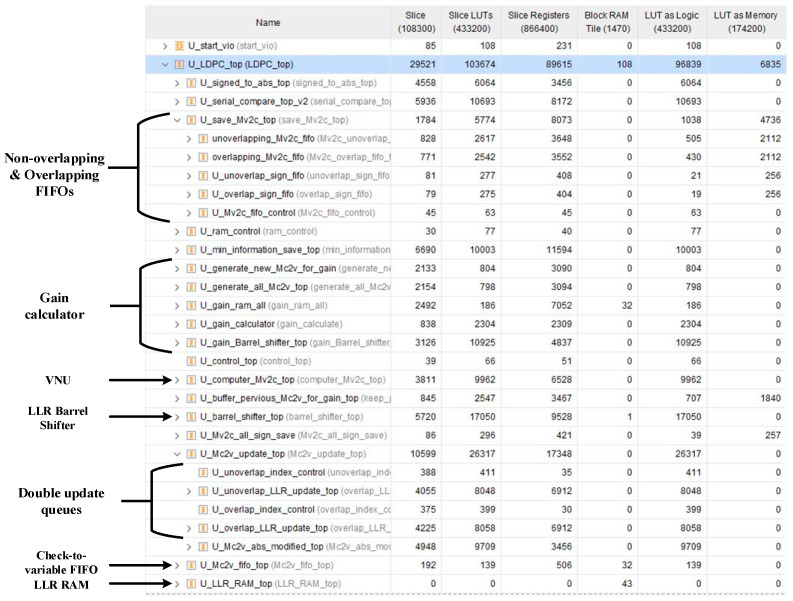
The resource usage of every module in the hardware implementation of priority-based decoder with double update queues.

**Table 1 sensors-22-03508-t001:** Implementation results for 5G NR and WiMAX decoders in comparison with previous works.

	This Work	[15]	[12]	This Work	[15]	[14]
Length	10,368 (code rate = 22/27)			2304 (code rate = 3/4)		
Standard	5G NR			WiMAX		
Device	xc7vx690t	xc7vx690t	xc7k160t	xc7vx690t	xc7vx690t	xc7vx485t
Quant	{8,8,6}	{8,8,6}	{5,8,6}	{8,8,6}	{8,8,6}	{4,4,4}
Algorithm	OMSA	OMSA	OMSA	OMSA	OMSA	/
Slice	29,521	30,824	/	7477	7906	12,496
LUT	103,674	100,929	74,373	26,744	24,228	40,700
FF	89,615	85,431	46,517	19,594	23,290	26,925
36k BRAM	108	136.5	198.5	27	33.5	40.5
*f_max_* [MHz]	255.0	261.0	160.0	310.0	314.6	142.8
*T_norm_* [Gbps]	31.4	31.7	11.96	8.2	8.5	10.8

**Table 2 sensors-22-03508-t002:** Throughput ratio between priority-based layered decoder with double update queues and hybrid layered decoder [15].

	This Work			[15]			
SNR [dB]	*T_norm_* [Gbps]	*AIN*	*T* [Gbps]	*T_norm_* [Gbps]	*AIN*	*T* [Gbps]	*TR* [%]
5.9	31.4	11	2.85	31.7	30	1.0	285
6.0		9	3.5		21	1.5	233
6.1		8	3.93		16	2.0	197
6.2		7	4.5		14	2.3	196
6.3		6	5.2		12	2.6	200
6.4		6	5.2		11	2.9	179
6.5		6	5.2		10	3.2	163
6.6		5	6.3		9	3.5	180
6.7		5	6.3		8	4.0	158
6.8		5	6.3		8	4.0	158
6.9		4	7.9		8	4.0	198
7.0		4	7.9		7	4.5	176

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
