# Peer review of "High Throughput Priority-Based Layered QC-LDPC Decoder with Double Update Queues for Mitigating Pipeline Conflicts"

_sensors, 2022, doi:10.3390/s22093508_

Round 1
Reviewer 1 Report
This paper considers a high throughput layered decoder for quasi-cyclic (QC) low-density parity-check 10 (LDPC) codes. Particularly, the paper focuses on mitigating pipeline conflicts at a high operating frequency in a layered decoding schedule. Overall, the paper is well written, and the considered issue is interesting. In addition, the paper also makes a comparison over the state-of-the-art. This Reviewer, therefore, recommends this work for publication.
Reviewer 2 Report
In this paper, a novel priority-based layered schedule is proposed, with the aim to keep the updates of LLRs as frequent as possible when pipeline conflicts occur. Also, the application of double update queries is proposed.
The general idea is interesting, the background theory is mostly correctly presented in section 2.1. The novel results are presented in part in Subsection 2.2, and mostly in Section 3. Figures 1, 2, 3, and 4 are illustrative and present the main contribution of the paper.
However, I believe that the structure of the paper can be improved if two subsections are added in Section 1, namely 1.1. Related work and 1.2. Overview and contribution. The paragraph between lines 48 and 61 can be the base for section 1.1. Although the most important papers are already cited, maybe some additional papers should be discussed in this part (e.g. papers available from https://www.mdpi.com/2079-9292/10/9/1106/htm, https://www.mdpi.com/2079-9292/10/16/2010/pdf). Lines 62-75 can be used as the base for subsection 1.2, and here, the authors can give a short overview of the next sections, and emphasize the main contributions of the paper. In Section 2 you can present the known results, and the novel techniques can be presented in Section 3.
FER curves are presented for one LDPC code only - a 5G code with a rate 22/27. Based on the presented results, it is not easy to conclude that the observed effects stand for the other QC LDPC codes. I understand that the time period for preparing the revised manuscript is limited, but it is highly desirable to present the results for at least one additional code. It can be shorter, with a length of the codeword 1000-1200 bits, as WiMax or WiFi code.
When you state that one million Monte Carlo trials are undertaken, does it mean that one million codewords are sent? If it is, please add the comment about FER values that can be reliably estimated for this simulation parameters. FER(SNR) figures, presented in Figure 7 should be enlarged as the details are not visible when printed.
Please, check the mathematical expressions one more time. What is maximized and what is minimized in Eq. 5?
Please, rephrase line 384: "...additional iteration numbers are significantly reduced". When you observe sentence in lines 373-375, it is not clear what cases are comapered (0.4dB loss vs. 0.2 dB loss).
Also, there are some minor mistakes that have to be corrected before the publication:
- in line 78, please replace "messages-passing" with "message-passing"
- in Eqs. (1), (3), (7),... add a comma after the equation
- in Eqs. (2) and (5) place a dot after the equation, as the end of the sentence is the equation itself.
- in line 81, it is not usual to say "a posterior possibility", it is usually used the phrase "a posterior probability" even in the case when the quantity is related to the likelihoods.
- in line 98 replace "decoding degradations" with "decoding performance degradations"
- is the syndrome equal to zero the only condition for the decoding termination? Is the number of iterations unlimited? Please, add the additional condition in line 115.
- in references [3] and [4] the corresponding years should be added.
Reviewer 3 Report
Hardware acceleration is considered for LDPC codes. I like the approach and the extensive implementation results - practical and impactful. The only comment I have is on the theoretical side - it would be great if a more detailed analysis on the conflict probability can be obtained. This will give further insights on if more than double queues would be helpful (e.g., three queues?) and how to adjust the queue size in a more adaptive manner (to further save resource). Overall, pipeline conflict appears an important and relevant problem that shall be addressed. In addition to the experimental based approach adopted in this work, theoretical analysis could also shed light, which is somewhat missing now.
Round 2
Reviewer 2 Report
Most of my remarks are taken into account, and all mistakes are corrected.
Please, avoid placing only the title of the section/subsection at the end of the page, if the text starts at the top of the next page. Also, the font size in figure 1 should be reduced.
These corrections can be made in the phase of the proof preparation, and it is not necessary to extend the review process.